# The Use of the Nitroblue Tetrazolium Test in Blood Granulocytes for Discriminating Bacterial and Non-Bacterial Neutrophilic Dermatitis

**DOI:** 10.3390/vetsci11120634

**Published:** 2024-12-07

**Authors:** Marina García, Icíar Martínez-Flórez, Laia Solano-Gallego, Nuria García, Laura Ordeix

**Affiliations:** 1Fundació Hospital Clínic Veterinari, Universitat Autὸnoma de Barcelona, 08193 Bellaterra, Spain; 2Departament de Medicina i Cirurgia Animals, Universitat Autὸnoma de Barcelona, 08193 Bellaterra, Spain

**Keywords:** NBT, dog, neutrophil, superficial pyoderma, sterile neutrophilic dermatitis, oxidative metabolism

## Abstract

The nitroblue tetrazolium (NBT) reduction test measures metabolic activity in cells. This study aimed to compare reactive oxygen species (ROS) production in neutrophils from healthy dogs, dogs with superficial pyoderma, and dogs with sterile neutrophilic pustular dermatitis. Twenty-eight dogs were divided into three groups, and their blood was tested using the NBT reduction method. The results showed that dogs with sterile neutrophilic dermatitis had higher NBT rates, especially if they tested positive for *Leishmania* spp. The NBT test may help identify systemic neutrophil activation in immune-mediated diseases, but it was less effective in distinguishing between healthy dogs and those with superficial pyoderma.

## 1. Introduction

The nitroblue tetrazolium (NBT) reduction assay is commonly used to assess metabolic activity in both mammalian and microbial cells [1,2]. First introduced in 1968, it initially served as a key diagnostic tool in human medicine for chronic granulomatous disease in infants [3]. Over time, its applications in human medicine have expanded to include the assessment of various inflammatory and infectious conditions, such as viral meningoencephalitis [4], tuberculosis [5], hepatic abscesses caused by amebiasis [6], and several skin diseases [7]. Additionally, the test has been employed as a general indicator of infection risk [8].

The NBT reduction test is designed to indirectly assess the release of reactive oxygen species (ROS) by neutrophils by measuring their NADPH oxidase activity [9]. During the process, NBT, which is initially soluble and colorless, is reduced by NADPH oxidase in active neutrophils within the phagosome, converting into formazan, an insoluble substance with a greyish-blue color. The rate at which NBT is reduced is assessed by calculating the percentage of neutrophils that contain formazan in the cytoplasm through conventional optical microscopy. The level of NBT reduction correlates directly with the amount of ROS generated by phagocytes during the oxidative burst [10]. Thus, the NBT reduction test serves as a method to indirectly evaluate ROS release by neutrophils by measuring NADPH oxidase activity [9].

In veterinary practice, the NBT reduction test has been used to evaluate neutrophil activity in various conditions, including canine leishmaniosis [2,11], ehrlichiosis [12], diabetes mellitus [13], and transient immunosuppression following polyvalent vaccination [14]. In a study on ehrlichiosis, although no significant differences were observed in the NBT test between infected and healthy dogs, neutrophils from infected dogs showed higher NBT-reducing activity after stimulation, suggesting an active response during the acute phase of the disease [12]. In studies on leishmaniosis, neutrophil oxidative activity was found to be higher in the early stages of infection compared with seropositive healthy dogs and in the advanced stages of the disease [2,11]. In dogs with diabetes mellitus, another study showed an increase in NBT reduction, which decreased after stimulation, possibly indicating a reduced bactericidal capacity of neutrophils [13]. Finally, one study reported a significant decrease in neutrophil activity following polyvalent vaccination, suggesting a transient immunosuppression associated with a regulatory shift towards a TH2-type immune response [14]. Although the NBT has been studied in a variety of veterinary conditions, no studies have investigated its use in the diagnosis of bacterial skin infections in dogs.

In veterinary dermatology, several diseases may exhibit similar clinical patterns. For example, superficial pyoderma and immune-mediated diseases like sterile neutrophilic dermatitis can present as papulopustular dermatitis [15,16]. Therefore, for an accurate diagnosis, complex clinical reasoning with the implementation of several diagnostic tests. Such as cutaneous cytology, dermatopathological examination, bacterial cultures, and even response to specific antibiotic treatment, is needed [15]. Due to the challenges associated with the diagnosis of this clinical pattern, the initial hypothesis for this study was that the NBT reduction test may be a valuable tool for differentiating between certain bacterial infections from non-bacterial diseases. Therefore, the objective was to evaluate and compare the differences in ROS production by peripheral blood neutrophils in healthy dogs, dogs with superficial pyoderma, and those with sterile neutrophilic dermatitis.

## 2. Materials and Methods

### 2.1. Study Animals

Eighteen dogs of different breeds were selected from patients attending the dermatological service at the veterinary hospital of Universitat Autònoma de Barcelona (Fundació Hospital Clínic Veterinari). These dogs were retrospectively divided into two groups. Group 1 included ten dogs with superficial pyoderma diagnosed based on clinical signs (papulopustular dermatitis), skin cytology (neutrophilic inflammation with intracellular cocci), and a favorable response to antibiotic treatment. Group 2 included eight dogs with sterile neutrophilic dermatitis. This diagnosis was based on clinical signs (papulopustular or placo-nodular dermatitis), skin cytology (neutrophilic inflammation in the absence of bacteria), and histopathological examination (neutrophilic pustular dermatitis with or without folliculitis) and confirmed through negative bacterial cultures or lack of response to specific antibiotic treatment and response to immunomodulatory treatment. Subgroup 2A consisted of three dogs with sterile neutrophilic pustular dermatitis, all seropositive for *Leishmania* spp. Two of the dogs were in stage IIa and one was in stage IIb according to the LeishVet stage [17] classification. Serum protein electrophoresis revealed an increase in gamma globulins. The urine protein-to-creatinine ratio (UPC) was <0.5 in two cases (non-proteinuric) and between 0.5 and 1 in one case (proteinuric). The ELISA for *Leishmania infantum* antibodies was highly positive (≥300 EU) in all three dogs. Subgroup 2B included five dogs with sterile neutrophilic dermatitis that were seronegative for *Leishmania* spp. Moreover, ten healthy dogs were included in group 3 as control dogs. Group 3 consisted of residual samples obtained during preliminary analytical tests for castration and ovariohysterectomy.

### 2.2. Ethics

In this study, residual patient samples were used, and our analysis did not involve direct research on animal subjects. In addition, the owners of the patients gave their informed consent on the day of the initial hospital visit and authorized the use of the data from the samples collected for diagnostic purposes.

### 2.3. Blood Collection

Blood samples were collected from the cephalic or jugular veins and transferred to EDTA tubes for hematological analysis and NBT test prior to any antibacterial treatment in group 1 and systemic corticosteroid therapy and/or anti-*Leishmania* treatment in group 2.

### 2.4. Blood Nitroblue Tetrazolium Reduction Test

In this study, the chosen form was powder (N6876, Sigma-Aldrich Co., St. Louis, MO, USA), and it was prepared by diluting it in phosphate-buffered saline (PBS) solution to a concentration of 0.1%. The NBT test was performed as previously described. The procedure carried out was as follows [1]: 100 μL of blood was placed into separate Eppendorf tubes and left for 15 min at room temperature. The blood was gently mixed before filling three 40 mm/20 μL hematocrit capillary microtubes. These capillary tubes were centrifuged at 2910× *g* for 5 min to collect the buffy coat. The buffy coat from the three hematocrit tubes (5 μL total) was transferred into an Eppendorf containing an equal amount of 0.1% NBT solution. The Eppendorf was gently mixed and incubated at 37.5 °C for 15 min, then allowed to rest at room temperature for an additional 15 min. Two blood slides were prepared from each Eppendorf by placing 2 μL of the NBT-treated blood on each glass slide. To evaluate the NBT percentage, 100 neutrophils with distinct morphology were counted on each slide using standard light microscopy (Figure 1). Only one author (MG) performed all the examinations. The NBT percentage was calculated by determining the proportion of activated neutrophils, indicated by the presence of blue-black formazan deposits, among the total neutrophils counted.

### 2.5. Statistical Analysis

Statistical analysis of the data was performed with GraphPad Prism 8.0 for Mac (GraphPad Software, San Diego, CA, USA). A series of normality tests, including Anderson–Darling, D’Agostino and Pearson, Shapiro–Wilk, and Kolmogorov–Smirnov, were conducted to assess the normality of NBT rates. The NBT rates followed a normal distribution, and a *p*-value < 0.05 was considered statistically significant. For comparisons between the three groups, Welch’s ANOVA test was used. Results were considered statistically significant when the *p*-value was <0.05.

## 3. Results

A total of 28 dogs were included in the study. Group 1 consisted of ten dogs with superficial pyoderma, with a median age of 5.5 years [3 to 11 years]. Of these, five were neutered females, three were neutered males, and two were intact males. Regarding breed, seven dogs were purebred (Pincher, Dogue de Bordeaux, Yorkshire Terrier, American Staffordshire, Schnauzer, and Maltese Bichon), while three were mixed-breed. In all these patients the primary cause of superficial pyoderma was allergic dermatitis. Group 2 included eight dogs with sterile neutrophilic dermatitis, subdivided into two subgroups. Subgroup 2A consisted of three dogs with sterile neutrophilic dermatitis and leishmaniois, with a median age of 5 years [4 to 6 years]. Of these, one was an intact male and two were neutered females, all of mixed breed. Subgroup 2B comprised five dogs, with a median age of 5 years [4 to 6 years]. Of these, one was an intact female and four were neutered females. Regarding breed, one dog was mixed-breed and four were purebred (Podencos, Whippet, and Maltese Bichon). Group 3 consisted of ten healthy dogs, with a median age of 3 years [1–7 years]. This group included five intact males and five intact females. Of these, nine were purebred (Border Collie, Yorkshire Terrier, German Shepherd, Bull Terrier, Shetland Sheepdog, and two Beagles) and one was mixed-breed.

The median results of total leukocyte and neutrophil count are presented in Table 1. No statistically significant differences were observed in the total concentrations of leukocytes and neutrophils.

The mean and standard deviation of the NBT reduction test obtained in all the groups are shown in Table 2. The statistical analysis (Figure 2) revealed no statistically significant differences between group 1 and group 3 (*p* = 0.15). However, significant differences were identified between group 2 and group 3 (*p* = 0.01) as well as between group 1 and group 2 (*p* = 0.04). Moreover, within group 2, group 2A exhibited a significantly higher NBT rate compared with group 2B (*p* = 0.004).

## 4. Discussion

The objective of this study was to evaluate and compare the changes in the generation of ROS by neutrophils in the peripheral blood of healthy dogs, dogs with superficial pyoderma, and those with sterile neutrophilic dermatitis. Identifying a rapid and accurate indicator that can differentiate between a bacterial infection and a sterile condition would have significant clinical implications, facilitating therapeutic decision-making and reducing the risk of unnecessary diagnostic or therapeutic interventions [18].

Currently, there is no available information in the literature on the use of the NBT reduction test to diagnose any bacterial infection in dogs other than ehrlichiosis. However, in human medicine, this is a controversial issue as some studies have supported its use as an indicator of bacterial infections [18,19,20], whereas others have questioned its utility, arguing that the test measures a disease response that is not exclusive to bacterial infections and depends on the normal phagocytic function of neutrophils [21].

The rate of reduction in NBT that was observed in the healthy group was like that reported in previous studies [1,2,11]. Moreover, the results of the study presented here revealed no significant differences between healthy dogs and dogs with superficial pyoderma, suggesting that there is no systemic activation of neutrophils in cases of superficial pyoderma. Interestingly, differences were identified between dogs with sterile neutrophilic dermatitis and healthy dogs and dogs with superficial pyoderma. These findings suggest the possibility of increased oxidative metabolism of neutrophils in the presence of a systemic immune-mediated condition. However, due to the limited number of cases in each group, further studies are necessary to verify this observation. Additionally, within the group with sterile neutrophilic dermatitis, dogs with leishmaniosis had a significantly higher NBT reduction test rate compared with seronegative dogs, suggesting that neutrophil activation is closely related to *Leishmania* infection. According to the reviewed literature, the increase in neutrophil oxidative metabolism is more pronounced in dogs with a mild stage (I LeishVet [17]) and healthy seropositive dogs than in dogs in more advanced stages of the disease (II-III-IV LeishVet [17]) [2,11]. Although clinical staging of dogs with leishmaniosis was not an initial objective, it is interesting to note that they were highly seropositive. The dogs in our study were at an advanced stage of leishmaniosis (stage IIa and stage IIb LeishVet [17]). This difference from previously published data could be due to the combination of systemic immune-mediated disease and leishmaniosis.

There are some limitations in the present study. The main one is the small number of dogs included. To confirm the usefulness of the NBT reduction test to discriminate bacterial and non-bacterial neutrophilic dermatitis, it would be necessary to increase the number of subjects studied, particularly in cases of sterile neutrophilic dermatitis without leishmaniosis. In addition, it would be very interesting to include patients with pemphigus foliaceous in this group, which is the most known purulent, immune-mediated skin disease in dogs. Moreover, it would be interesting to evaluate this test on peripheral neutrophils, especially in dogs with purulent skin diseases. However, although the authors attempted this experimental design, it was unsuccessful due to technical aspects (personal observation).

## 5. Conclusions

The results indicate that the NBT test does not appear to distinguish between healthy dogs and those with superficial pyoderma. However, dogs with sterile neutrophilic dermatitis showed significantly higher NBT rates, especially if they were seropositive for *Leishmania* spp., suggesting increased production of ROS. These findings suggest that the NBT test could be a useful tool for differentiating immune-mediated diseases such as sterile neutrophilic dermatitis, particularly when associated with *Leishmania* spp. infection.

## Figures and Tables

**Figure 1 vetsci-11-00634-f001:**
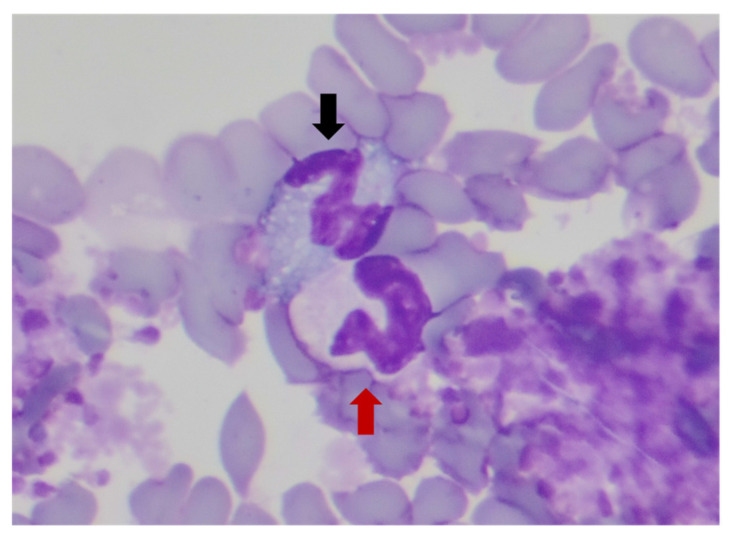
A neutrophil with formazan deposits in the cytoplasm (black arrow) and another without formazan accumulation in the cytoplasm (red arrow), stained with Diff-Quick™ (Sigma-Aldrich Co., ST. Louis, USA), magnification ×100.

**Figure 2 vetsci-11-00634-f002:**
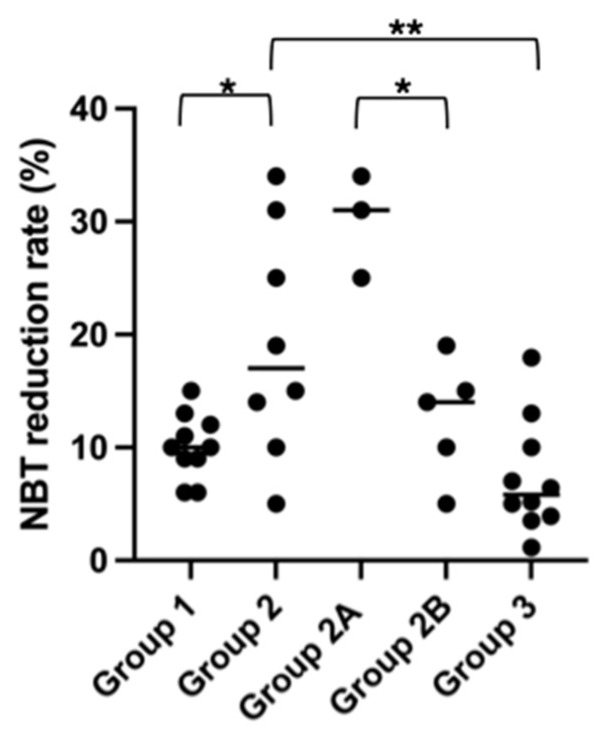
Nitroblue tetrazolium reduction test rate in the different groups studied. NBT = nitroblue tetrazolium. Group 1: Superficial pyoderma. Group 2: Sterile neutrophilic dermatitis. Group 2A: Sterile neutrophilic dermatitis and leishmaniosis. Group 2B: Sterile neutrophilic dermatitis without leishmaniosis. Group 3: Healthy dogs. (* *p* < 0.05, ** *p* < 0.01).

**Table 1 vetsci-11-00634-t001:** Median results for total leukocyte and neutrophil count. Group 1: Superficial pyoderma. Group 2: Sterile neutrophilic dermatitis. Group 2A: Sterile neutrophilic dermatitis and leishmaniosis. Group 2B: Sterile neutrophilic dermatitis without leishmaniosis. Group 3: Healthy dogs.

Median (min–max)	Group 1	Group 2	Group 2A	Group 2B	Group 3
Leukocytes (cell/µL)	9415 (4430–12,720)	12,470 (4430–22,150)	12,280 (6540–22,150)	12,660 (4430–18,800)	11,330 (6390–13,680)
Neutrophils (cell/µL)	6150 (3132–8432)	8939 (3132–18,429)	9730 (4898–18,429)	5426 (3132–15,330)	6661 (3706–9644)

**Table 2 vetsci-11-00634-t002:** Nitroblue tetrazolium reduction test (%) rates (mean +/− standard deviation). Group 1: Superficial pyoderma. Group 2: Sterile neutrophilic dermatitis. Group 2A: Sterile neutrophilic dermatitis and leishmaniosis. Group 2B: Sterile neutrophilic dermatitis without leishmaniosis. Group 3: Healthy dogs.

Group 1	Group 2	Group 2A	Group 2B	Group 3
10 ± 2.8%	19 ± 10%	30 ± 4.6%	13 ± 5.3%	7.3 ± 5%

## Data Availability

Data is contained within the article.

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
