# Peer review of "The Use of the Nitroblue Tetrazolium Test in Blood Granulocytes for Discriminating Bacterial and Non-Bacterial Neutrophilic Dermatitis"

_vetsci, 2024, doi:10.3390/vetsci11120634_

Round 1

Reviewer 1 Report

Comments and Suggestions for Authors

Garcia et al. proposed rather short communication than an article on the use of nitroblue tetrazolium in dermatology. Since this method is well known and evaluated in human medicine and veterinary, its’ innovative perspective is limited, however it provided interesting results. I have some comments on the quality of the work listed below.

General comments:

The references do not follow the journal guidelines; please correct citations in the text and bibliography.

Introduction:

It is quite short, it would be useful to indicate how the evaluation of formazan is made. A brief description of results in other diseases from lines 57-61 would be great for the manuscript.

Materials and methods:

The manuscript addressing of ethical issues is very little. Was the approval of the local committee acquired? If those were patients, information on a formal agreement to enroll in the study from the owners had to be collected and stated in the article.

The description of patients is scarce, it only involves median age and sex, which in such small group as 28 individuals should provide more details (race, co-morbidities, etc)

If I understood correctly, the values in square brackets are the age of the dogs? (in lines 80, 85)

Were all the animals treated during enrollment? Line 88-89 confirmed through negative bacterial cultures or lack of response to specific antibiotic treatment and response to immunomodulatory treatment.

Both treatments can change the neutrophil's answer in organisms, thus influencing the result of NBT test, please, explain, if the treatment was continued, and what treatment it was for each animal (summarizing table would help greatly)

The number of animals is not correct; from the description of groups 1-3 is 33, not 28.

Was the NBT test performed by one person or more?

What kind of ANOVA was used?

Results:

The table caption has to be over the table.

Figure 2. The significant differences should be indicated on the graph (e.g., with asterisks, symbols, etc.)

Generally, one result is too little to be considered for publication. There is no clinical evaluation of the patient; there were some biochemistry results (I recall the protein/creatinine ratio, was the CBC also performed?) It would improve the readability  and presentation of results.

Was the NBT followed in the future? Was it performed a second time after successful treatment?

I see from credit that the method was validated; how was the validation performed?

Conclusions

The last sentence is unnecessary and should be removed

Author Response

The authors greatly appreciate the comments from Reviewer 1, as well as her/his thorough review of the manuscript. We have carefully addressed all the suggested changes.

General comments:

Comment 1: The references do not follow the journal guidelines; please correct citations in the text and bibliography.

Response 1: We are very sorry for the mistake. The References section has been revised according to the citation style recommended by Veterinary Science (ACS reference style).

Introduction:

Comment 2: It is quite short; it would be useful to indicate how the evaluation of formazan is made. A brief description of results in other diseases from lines 57-61 would be great for the manuscript.

Response 2: The suggested recommendations have been incorporated into the new version of the article. (lines 51-54 and 58-73).

Materials and methods:

Comment 3: The manuscript addressing of ethical issues is very little. Was the approval of the local committee acquired? If those were patients, information on a formal agreement to enroll in the study from the owners had to be collected and stated in the article.

Response 3: Ethical study approval was not needed due to the use of residual blood samples in this study. Moreover, our analysis did not involve direct research with animal subjects. Additionally, on the day of the first visit in our hospital, the owners of the patients sign an informed consent form and provide authorization for the transfer of data derived from the samples collected for diagnostic purposes for scientific research reasons. This document has been submitted to the journal. (Lines 112-117).

Comment 4: The description of patients is scarce, it only involves median age and sex, which in such small group as 28 individuals should provide more details (race, co-morbidities, etc).

Response 4: The description of the patients has been expanded and can now be found in the results section. (Lines 169-184).

Comment 5: If I understood correctly, the values in square brackets are the age of the dogs? (in lines 80, 85)

Response 5: Yes, the age is provided with median and then the range is provided in brackets.

Comment 6: Were all the animals treated during enrollment? Line 88-89 confirmed through negative bacterial cultures or lack of response to specific antibiotic treatment and response to immunomodulatory treatment. Both treatments can change the neutrophil's answer in organisms, thus influencing the result of NBT test, please, explain, if the treatment was continued, and what treatment it was for each animal (summarizing table would help greatly).

Response 6: The NBT test was only done on day 0, before antibacterial treatment in group 1 and before systemic corticosteroid therapy and/or anti-Leishmania treatment in group 2. It would have been very interesting to do it after treatment to see if there were any changes, but this was not considered at the time. It has been clarified in the manuscript. (Lines 120-123).

Comment 7: The number of animals is not correct; from the description of groups 1-3 is 33, not 28.

Response 7: Apologies for the error. In total, there are 28 dogs, of which 10 were healthy dogs (Group 3) and eighteen were diseased dogs (Group 1 consists of 10 dogs with superficial pyoderma) and Group 2 includes 8 dogs with sterile neutrophilic dermatitis. This has been corrected in the group description. (Lines 89-109).

Comment 8: Was the NBT test performed by one person or more?

Response 8: The NBT test was performed by a single individual, particularly de first author. This has been clarified in line 139-140 of the new version.

Comment 9: What kind of ANOVA was used?

Response 9: Welch ANOVA test. This has been specified in line 166.

Results:

Comment 10: The table caption must be over the table.

Response 10: The text above the table has been updated. (Lines 207 -210).

Comment 11: Figure 2. The significant differences should be indicated on the graph (e.g., with asterisks, symbols, etc.)

Response 11: Significant differences have been indicated in the chart using asterisks.

Comment 12: Generally, one result is too little to be considered for publication. There is no clinical evaluation of the patient; there were some biochemistry results (I recall the protein/creatinine ratio, was the CBC also performed?) It would improve the readability and presentation of results.

Response 12: Thank you for your comment. We agree with your comment and have incorporated it into the results section. (Lines 186-199).

Comment 13: Was the NBT followed in the future? Was it performed a second time after successful treatment?

Response 13: Unfortunately, the NBT test was not performed after treatment in this study. However, it would be very interesting to include a post-treatment NBT test in future studies to evaluate potential changes and gain a more comprehensive understanding.

Comment 14: I see from credit that the method was validated; how was the validation performed?

Response 14: Apologies for the confusion; there has been a misunderstanding. The test has been performed in accordance with the previously described procedure.

Blasi-Brugué, C.; Martínez-Flórez, I.; Baxarias, M.; del Rio-Velasco, J.; Solano-Gallego, L. Exploring the Relationship between Neutrophil Activation and Different States of Canine L. infantum Infection: Nitroblue Tetrazolium Test and IFN-γ. Vet. Sci 2023, 10 (9).

Gómez-Ochoa, P.; Lara, A.; Couto, G.; Marcen, J. M.; Peris, A.; Gascón, M.; Castillo, J. A. The Nitroblue Tetrazolium Reduction Test in Canine Leishmaniosis. Vet. Parasitol 2010, 172, 135–138.

Conclusions

Comment 15: The last sentence is unnecessary and should be removed

Response 15: It has been removed from the text.

Many thanks for your attention and help, 

Sincerely yours.

Reviewer 2 Report

Comments and Suggestions for Authors

The article aims to understand if Nitroblue Tetrazolium Test can be useful in identifying the presence of bacterial superficial dermatitis in dogs. 

The article is really well written, linear, schematic, easy to read, with a clear objective and clear conclusions and for that I compliment the authors.

The information that comes from it adds a piece to the knowledge on the subject and provides ideas for further studies on the subject. I therefore believe that it is absolutely ready to be published.

Below I report some minor suggestions:

line 77: which dog breeds were represented in the three groups?

line 81: are the youngest and oldest ages of the group reported in square brackets? please specify this the first time this writing method is used in the text.

line 97: I believe there is a repetition in the sentence, I would remove five dogs and change it to the following: ..."which included five female dogs with median age was 5 years...."

line 99: why did the healthy ones come to the visit?

line 100: I don't understand the median age of 1 year with extremes 1-7. I believe there is an error in the reported average number. If it were confirmed that the average is 1 year, then I believe that within the limits of the study it should be reported that in the control group the healthy subjects were decidedly younger than the two groups of sick subjects and this should be discussed as a possible factor influencing the results.

line 106: to evaluate what was taken from the healthy subjects, given that the authors say that surplus blood was used?

Author Response

The authors greatly appreciate the comments from Reviewer 2, as well as her/his thorough review of the manuscript. We have carefully addressed all the suggested changes.

Comment 1: line 77: which dog breeds were represented in the three groups?

Response 1: The description of the patients has been expanded and can now be found in the results section. (Lines 169-184).

Comment 2: line 81: are the youngest and oldest ages of the group reported in square brackets? please specify this the first time this writing method is used in the text.

Response 2: Yes, the age is provided with median and then the range is provided in brackets. Thanks for the suggestion, it has been modified in the manuscript.

Comment 3: line 97: I believe there is a repetition in the sentence, I would remove five dogs and change it to the following: ..."which included five female dogs with median age was 5 years...."

Response 3: Thank you, it was a mistake, it has already been modified in the manuscript.

Comment 4: line 99: why did the healthy ones come to the visit?

Response 4: This question has been clarified in the text (lines 108-110). The dogs in the control group (Group 3) visited the hospital for a preliminary castration appointment. During this visit, a complete blood count and biochemical analysis were performed to evaluate their suitability for anesthesia. The residual blood used in the study was obtained from these tests.

Comment 5: line 100: I don't understand the median age of 1 year with extremes 1-7. I believe there is an error in the reported average number. If it were confirmed that the average is 1 year, then I believe that within the limits of the study it should be reported that in the control group the healthy subjects were decidedly younger than the two groups of sick subjects and this should be discussed as a possible factor influencing the results.

Response 5: Thank you for your comment. I would like to clarify that the reported median age of 1 year was an error. The correct median age is 3 years.

Comment 6: line 106: to evaluate what was taken from the healthy subjects, given that the authors say that surplus blood was used?

Response 6: This question has been addressed in the response to Comment 4. (lines 108-110).

Many thanks for your attention and help, 

Sincerely yours.

Reviewer 3 Report

Comments and Suggestions for Authors

Interesting study. The authors should expand the number of cases included to clarify whether the difference between the sterile neutrophilic dermatosis group and the superficial pyoderma one is clearly significant. A further study could include dogs with pemphigus foliaceus (the most common autoimmune disease in dogs).

Author Response

The authors sincerely acknowledge and understand the comments of Reviewer 3 and greatly appreciate her/his revision.

Comment 1: This value is not clear to which comparison of groups it refers. The first with the second, the second with the third or the first with the third??!

Response 1: Thank you very much for your comment. The text of the abstract has been revised for clarity. (Lines 23-26).

Comment 2: Are these seronegative dogs healthy animals or with sterile neutrophilic dermatitis?

Response 2: Thank you for your comment. They were dogs with sterile neutrophilic dermatitis and seronegative for Leishmania spp., the manuscript abstract has been modified for clarity. (Line 27-28).

Comment 3: Immune-mediated skin diseases in dogs have pathogenetic mechanisms that are not always the same and considering sterile neutrophilic dermatosis is quite rare in dogs, this generalization of results is questionable.

Response 3: Thank you for your comment, we completely agree. The manuscript abstract has been modified. (Lines 28-30).

Comment 4: Actually, the most important comparison for the study (G1-G2) has a p<0.041 which is very close to the value, decided by the authors, of statistically significant difference (p<0.05). Considering the low number of cases included in the different groups, this conclusion must be verified with further studies. It is suggested to include and comment on this criticality of the results in the discussion.

Response 4: Thank you very much for your comment. We have addressed this comment in the discussion section (Lines 252-256).

Comment 5: The authors may include examples such as pemphigus foliaceus, which is the most common purulent, immune-mediated skin disease in dogs.

Response 5: We completely agree with your comment. We coincide in that it is the most know immunomediated cutaneous disease in terms of etiology and in that it would be very interesting to include patients with this disease in this group. We have included a sentence in the discussion section. (Lines 272-274)

Many thanks for your attention and help, 

Sincerely yours.

Reviewer 4 Report

Comments and Suggestions for Authors

See attached file.

Author Response

The authors are very thankful for the revisions and sincerely acknowledge and understand the comments of Reviewer 4.

Comment 1: Lines 154-155, in my opinion, the ANOVA analysis is statistically unreliable for comparison among groups with eight subjects, such as Group 2. I would recommend increasing the number of dogs in Group 2, if possible, or using another statistical analysis test (no ANOVA).

Response 1: Thank you for your comment. We agree with your observation. The sample size in group 2 is small due to the rarity of cases of sterile neutrophilic dermatitis in clinical practice. We acknowledge that it would be ideal to increase the sample size in future studies. We would like to clarify that Welch ANOVA was used for statistical analysis, as this is appropriate in situations where sample sizes are unequal or where there are heterogeneities in the variances between groups. In addition, a normality test was performed on the data prior to analysis, which confirmed the use of Welch ANOVA.

Many thanks for your attention and help, 

Sincerely yours.

Round 2

Reviewer 1 Report

Comments and Suggestions for Authors

Dear authors,

Great work on revising your manuscript, it's readability has improved and I can recommend it after small changes.

Line 116: Last sentence should be deleted, it is redundant

Table. 1 – Please consider not using comma separation of thousands

And in Data availability the statement: Not applicable is used when no new data was generated and the article says otherwise. Consider adding "Data is contained within the article" Maybe by adding results of data from Table 2 as a supplementary material

Author Response

The authors greatly appreciate the comments from Reviewer 1.

Comment 1: Line 116: Last sentence should be deleted, it is redundant

Response 1: Thank you very much for your observation. We have removed the sentence from the manuscript.

Comment 2: Table. 1 – Please consider not using comma separation of thousands

Response 2: We appreciate your comment. The thousands comma has been removed from the table.

Comment 3: And in Data availability the statement: Not applicable is used when no new data was generated and the article says otherwise. Consider adding "Data is contained within the article" Maybe by adding results of data from Table 2 as a supplementary material.

Response 3: Thank you very much for your suggestion. We have updated the data availability statement to: "Data is contained within the article". (Line 292).

Many thanks for your attention and help.

Reviewer 4 Report

Comments and Suggestions for Authors

Respected Authors, I am sorry, but, in my opinion, your manuscript is still poor in results and content, needs additional experiments, has an insufficient number of References, etc.

Author Response

The authors appreciate the comments from Reviewer 4.

Comment 1: Respected Authors, I am sorry, but, in my opinion, your manuscript is still poor in results and content, needs additional experiments, has an insufficient number of References, etc.

Response 2: Thank you for your thoughtful comments and for taking the time to review our manuscript again. We greatly appreciate your perspective. While we respect your opinion, we believe that the results, content, and references presented in the manuscript provide a solid foundation for the conclusions drawn. To our knowledge, there are currently no studies in veterinary dermatology regarding the use of the NBT test. We hope that this article will serve as a valuable starting point for future research in this area.

Many thanks for your attention and help.